# Research on the Performance of Diamond-Like Carbon Coatings on Cutting Aluminum Alloy: Cutting Experiments and First-Principles Calculations

**Biao Huang [1], Er-geng Zhang [1,\*], Qiong Zhou [1,\*], Rong-chuan Lin [2] and Hao-ming Du [3]**

[1] Shanghai Engineering Research Center of Physical Vapor Deposition (PVD) Superhard Coating and Equipment, Shanghai Institute of Technology, Shanghai 201418, China; sit_bhuang@163.com

[2] School of Mechanical and Energy Engineering, Jimei University, Xiamen 361021, China; 13606033316@139.com

[3] School of Materials Science and Engineering, Shanghai Dianji University, Shanghai 200240, China; duhaom@163.com

\* Correspondence: zhangeg@yeah.net (E.-g.Z.); zhouqiong@sit.edu.cn (Q.Z.); Tel.: +86-21-60873356 (E.-g.Z. & Q.Z.)

**Abstract:** The purpose of this study is to investigate the cutting performance of amorphous carbon (a-C) coatings and hydrogenated amorphous carbon (a-C:H) coatings on machining 2A50 aluminum alloy. First-principles molecular dynamics simulation was applied to investigate the effect of hydrogen on the interaction between coatings and workpiece. The cross-section topography and internal structure of a-C and a-C:H films were characterized by field emission scanning electron microscopy and Raman spectroscopy. The surface roughness of the deposited films and processed workpiece were measured using a white light interferometer. The results show that the a-C-coated tool had the highest service life of 121 m and the best workpiece surface quality ($S_q$ parameter of 0.23 μm) while the workpiece surface roughness $S_q$ parameter was 0.35 and 0.52 μm when machined by the a-C:H-coated and the uncoated tool, respectively. Meanwhile, the build-up edge was observed on the a-C:H-coated tool and a layer of aluminum alloy was observed to have adhered to the surface of the uncoated tool at its stable stage. An interface model that examined the interactions between H-terminated diamond (111)/Al(111) surfaces revealed that H atoms would move laterally with the action of cutting heat (549 K) and increase the interaction between a-C:H and Al surfaces; therefore, Al was prone to adhere to the a-C:H-coated tool surface. The a-C coating shows better performance on cutting aluminum alloy than the a-C:H coating.

**Keywords:** amorphous carbon; hydrogenated amorphous carbon; coating; first-principles molecular dynamics; machinability; aluminum alloy

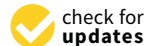



## 1. Introduction

Lightweight materials such as aluminum alloys, of which the strength to weight ratio is superior to that of steel, have been widely applied on automotive parts to reduce their weight [1–4]. However, the surface roughness of aluminum alloy has a significant influence on the performance of mechanical parts as well as production cost [5]. With the increasing requirement for enterprises to reduce production costs, it is necessary to improve the processing quality of aluminum alloy to meet the higher demand for these materials. The application of low friction and anti-adhesion coatings on cutting tools for machining aluminum alloys is one of the most widely used strategies to achieve low surface roughness.

The deposition of coatings on cutting tools is becoming popular because they have shown excellent chemical inertness, anti-adhesion, friction resistance, and self-lubrication properties. Hydrogenated amorphous carbon (a-C:H) is such a coating and has been used extensively to form low surface roughness [6]. However, the low thermostability of a-C:H

coating is the main drawback for dry and high-speed cutting applications. Hydrogen atoms were desorbed from an a-C:H coating at about 300 °C in the studies by Robertson [7], Tallant et al. [8], and Pei et al. [9]; thus, the structure of an a-C:H coating changes above 300 °C. This is because the C–H bond vibrates violently and then breaks under the action of thermal radiation; hence, the coating loses its surface chemical inertness and the surface energy increases. Softy/lubricating coatings (ZrN + MoST, MoZrN + MoST, and CrTiAlN + MoST) are also used to improve the surface roughness in dry cutting [10,11]. However, few of them have practical application in dry cutting. Sugihara et al. [12] developed a cutting tool with a micro-stripe textured surface to ensure an excellent anti-adhesive property in both wet and dry cutting. Nizar et al. also proposed a technique to improve the anti-adhesion by controlling the micro-texture of the cutting tool [13]. The micro-texture could reduce the contact area during the cutting process and provide storage for the lubricant; therefore, the surface roughness could be reduced, but the costs would increase dramatically. Another way to improve the surface roughness of aluminum alloy is to add coolants to most cutting processes as a lubricant to reduce friction and help chip removal [14,15]. However, the use of coolants is harmful to the environment and human health.

In recent years, numerous studies have addressed the tribological behavior of an amorphous carbon (a-C) coating against aluminum. A low coefficient of friction ($\leq 0.16$) and low wear rates of the a-C coating were observed against aluminum in an atmospheric environment and the a-C coating exhibited great lubricity as well [16–19]. Moreover, the thermal stability of the a-C coating was 450 °C, which was predicted by first-principles calculation in an oxygen atmosphere [7,20]. These studies show that the a-C coating's thermal stability and tribological behavior against aluminum alloy are superior to that of the a-C:H coating.

In light of these findings, the surface roughness of the workpiece and service life of the a-C coating tool, the a-C:H coating tool, and the uncoated tool when dry cutting aluminum alloy are compared in this paper. The work of adhesion and interaction energy between the cutting tool and aluminum surface was estimated by the first-principles calculation. The predictions of atomistic simulations at the interfaces were analyzed together with the results of the cutting experiments to elucidate the effect of hydrogen on the performance of carbon coatings against aluminum alloy surfaces.

## 2. Experimental Details

### 2.1. Multilayer Films Deposition

The a-C film and a-C:H film were prepared on a cemented carbide milling tool (D8 $\times$ 75 mm$^2$) and a YG10-grade cemented carbide (16 $\times$ 16 $\times$ 2.5 mm$^3$) substrate. Before film deposition, a standard pretreatment must be followed to achieve good bonding performance. Initially, the substrates were sprayed with Al$_2$O$_3$ (220#–260#) under 1.5 Pa gas pressure for 2 min, then polished with a cotton wheel (3500 r/min) for 3 min, and cleaned using ethanol for 10 min and deionized water for 5 min. Later, the substrates were dried with nitrogen gas flow. The tungsten carbide samples were used to characterize the microstructures and thickness of the coatings. The a-C:H coatings were prepared by a C$_2$H$_2$ and Ar gas mixture in a plasma discharging system while the a-C coatings were prepared by applying the pulsed magnetron sputtering technique. In the a-C:H coatings deposition system, a single, high-purity (99.9%) titanium target was pre-sputtered for approximately 10 min to remove impurities and deposited onto the substrates for 15 min as an underlayer with a current of 120 A and bias of −80 V. Then, the Ti-C:H functional gradient layer was deposited with a Ti target current of 80 A and a C$_2$H$_2$ flow of 50 sccm for 10 min. Subsequently, the a-C:H layer was deposited with a C$_2$H$_2$ flow of 65 sccm for 30 min. In a pulsed magnetron sputtering system, one titanium target and one graphite target were used to prepare the a-C coating. High-purity argon gas with a flow rate of 35 sccm was supplied continuously to maintain the process pressure of 0.3 Pa. Firstly, the substrates were etched under the bias voltage of −500 V for 20 min, then the surface was activated

and the oxide layer on the surface was removed. Secondly, the Ti underlayer was deposited onto the substrates with the Ti target frequency of 15 Hz and a current of 3.1 A for 20 min. Thirdly, the Ti-C layer was deposited with a Ti target current of 3.1 A and a graphite target current of 1.2 A. This process continued for 6 min and the frequency of Ti/graphite was 15 Hz. Subsequently, the amorphous carbon layer was deposited by sputtering one graphite target at the same frequency of 15 Hz for 60 min with a current of 1.2 A.

### 2.2. Characterization Techniques

The thickness of films was analyzed using a field emission scanning electron microscope (FE-SEM) (FEI inspect f50, Thermo Fisher Scientific, Hillsboro, OR, USA). The surface morphologies of films and workpieces were evaluated by a white-light interferometer (Contour GT-K0, Bruker, Billerica, MA, USA), and the measurement was carried out at room temperature (22 ± 3 °C) and 35–50% RH. Raman spectroscopy (Thermo Fisher Scientific) was used to characterize the structural properties of the films and was equipped with a 432 nm wavelength laser source operating at 2 mW power. The G peak and D peak positions and the ratio of peak intensities were fitted with a Gaussian line shape; $I_D/I_G$ was considered an indicator of carbon $sp^3/sp^2$ structure. The cutting performance of a-C:H and a-C multilayer films was estimated by conducting a dry milling experiment with a CNC milling machine (VMC-1000II, Nantong, China). YG10 cemented carbide (WC 90%, Co 10%), coated with a-C:H and a-C multilayer films for comparison, was selected as the tool substrate with a diameter of 8 mm, a rank angle ($\gamma_o$) of 15°, a clearance angle ($\alpha_o$) of 20°, and a helix angle (β) of 30°. The milling material was 2A50 aluminum alloy. The cutting test was performed under the following conditions: the spindle speed was 3980 r/min, feed rate $f = 0.1$ mm/r, and axial depth of cutting $a_p = 2$ mm. In this work, the abnormal vibration is regarded as a criterion for judging whether the cutting tool is blunt. The cutting temperature was measured by infrared thermometers (PA10, Keller, Unterschleissheim, Germany) with a temperature range of 0 to 1000 °C, a response time of less than 30 ms, and a measurement error of less than 1%. The temperatures were recorded every five seconds. The infrared array was focused on a certain position of the rake face, which is generally a distance from the tool tip, to capture the maximum temperature.

## 3. Results

### 3.1. Morphology

The surface morphologies (6336 μm × 4752 μm) of the cemented carbide substrates, and the a-C and a-C:H coatings, obtained by a white-light interferometer are shown in Figure 1. As can be seen, the cemented carbide substrates present undulant and convex-shaped WC particles that give the surface roughness $S_q$ parameter of 30 nm. After amorphous carbon deposition, the influence of the surface morphology of the substrate on the surface morphology of the coating is weakened due to the amorphous nature of its internal structure. In addition, the coating itself has no preferential orientation; therefore, no obvious hump particles are observed on the coating in this study, which brings about a relatively smooth surface $S_q$ parameter of 23 nm.

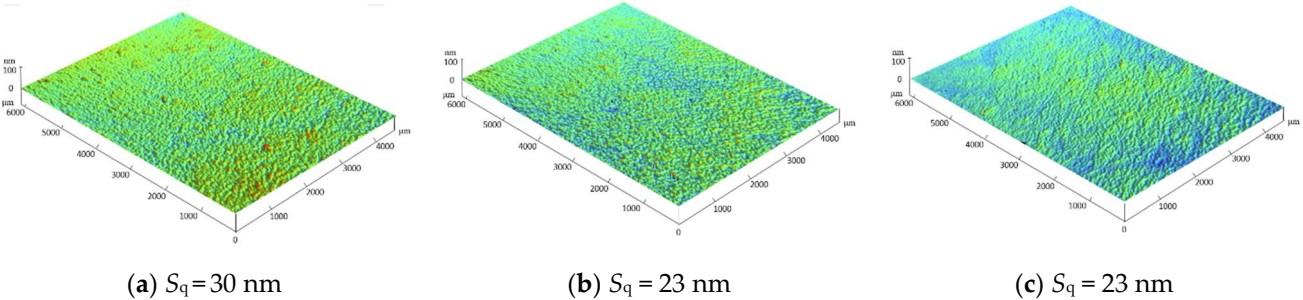

　　　　(**a**) $S_q$ = 30 nm　　　　　　　　　　　(**b**) $S_q$ = 23 nm　　　　　　　　　　　(**c**) $S_q$ = 23 nm

**Figure 1.** The surface morphologies: (**a**) YG10 substrates, (**b**) amorphous carbon (a-C) coating, (**c**) hydrogenated amorphous carbon (a-C:H) coating.

### 3.2. Cross-Section Topography

The thickness of the coating can be observed by a FE-SEM microscope. Figure 2 shows cross-section FE-SEM micrographs of the a-C:H and a-C films. The operating voltage is 10 KV. The thickness of both the a-C:H and a-C films is about 1.1 μm, the thickness of the Ti adhesive layer is around 0.3 μm, and the thickness of the Ti-C:H/Ti-C transition layer is about 0.1 μm. The Ti adhesion layer is closely connected with the substrate, which could improve the adhesion property of the coating. The Ti-C:H/Ti-C transition layer reduced the thermal expansion coefficient difference between the Ti adhesion layer and the a-C:H/a-C layer; thus, the internal stress of the coating is reduced. Meanwhile, the a-C:H/a-C layer presents in an amorphous state because no grain boundary is observed.

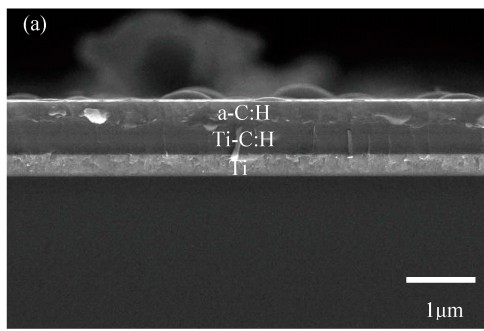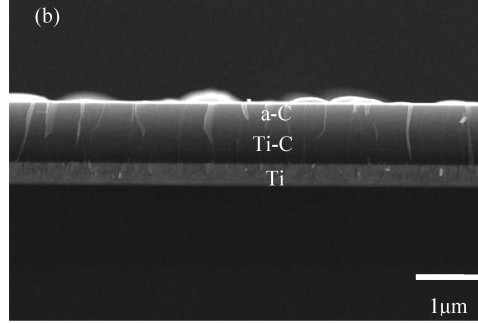

**Figure 2.** Cross-section field emission scanning electron microscope (FE-SEM) micrographs of (**a**) a-C:H, (**b**) a-C.

### 3.3. Microstructure

Raman spectra were utilized to estimate the $sp^2$ and $sp^3$ hybrid variations of the multilayer a-C:H and a-C films and are shown in Figure 3. Usually, thin amorphous carbon films exhibit two main Raman peaks, called the G peak, which is located in the vicinity of 1560 cm$^{-1}$, and the D peak, which is located around 1360 cm$^{-1}$, in the wavenumber region of 1000–1800 cm$^{-1}$ for visible excitation [21–25]. The G peak is due to the bond stretching of all pairs of $sp^2$ atoms in both rings and chains, whereas the D peak is the breathing mode of the rings [22]. The Raman spectra were deconvoluted using Gaussian peaks and the G peak and D peak were extracted. Table 1 shows the peak position and full width at half maximum (FWHM) of the D and G peaks and the intensity ratio of the D and G peaks ($I_D/I_G$ ratio) for the a-C:H and a-C samples. It can be seen that the D and G peak positions of the a-C:H film are shifted towards a high wavenumber compared with the a-C film. The $I_D/I_G$ ratio of the a-C:H sample is 0.74 and the $I_D/I_G$ ratio of the a-C sample is 0.93. Those behaviors are attributed to the decrease in the cross-linking degree and the enhanced $sp^3$ bonding in the structure [7]. Moreover, the FWHM of the G peak of the a-C:H film is 225 cm$^{-1}$, while that of the a-C film is 178 cm$^{-1}$. The FWHM of the G peak is related to the structure disorder (i.e., bond angles and bond lengths) and the amorphization degree of the films [21,26,27]. In other words, the H element in the a-C:H film could increase the FWHM of the G peak and result in the films being amorphous and disordered. Thus, the above results clearly reveal that the film with the H element presents higher $sp^3$ bonding and disorder.

**Table 1.** Peak position and full width at half maximum (FWHM) of D and G peaks and the intensity ratio of the D and G peaks ($I_D/I_G$ ratio) for deposited samples.

| Coatings | D-Peak Position (cm$^{-1}$) | D-Peak FWHM (cm$^{-1}$) | G-Peak Position (cm$^{-1}$) | G-Peak FWHM (cm$^{-1}$) | $I_D/I_G$ |
|---|---|---|---|---|---|
| a-C:H | 1390 | 361 | 1558 | 225 | 0.74 |
| a-C | 1368 | 306 | 1551 | 178 | 0.93 |

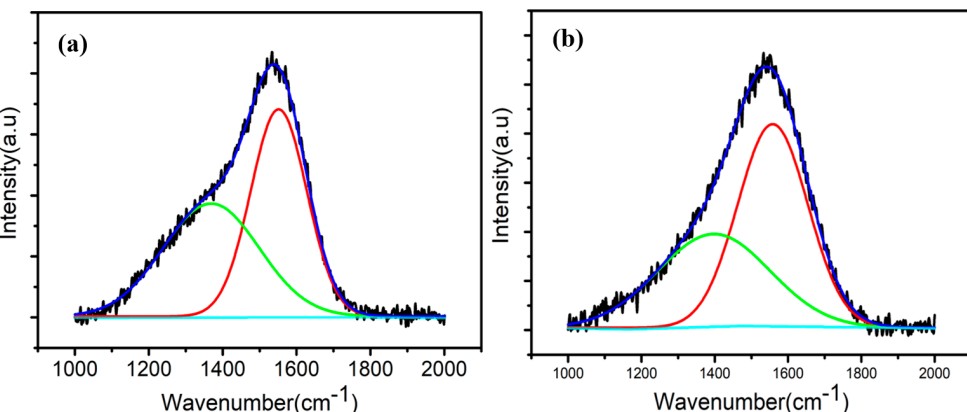

**Figure 3.** Raman spectra: (**a**) a-C:H samples, (**b**) a-C samples.

*3.4. Cutting Performance*

The cutting tests were conducted at a vertical CNC milling center. The surface quality of the workpiece, the machining distance of the milling tool, and the chip morphology were obtained by a dry milling experiment of aluminum alloy.

Figure 4 shows the relationship between the machining distance and machined surface roughness of uncoated, a-C:H-coated, and a-C-coated YG10 cemented carbide tools. The workpiece machined by the uncoated tool exhibits a relatively high surface roughness $S_q$ parameter of 1.02 μm at the stable wear stage with the machining distance of 20 m, and the tool failed at the machining distance of 38 m. The a-C:H and a-C-coated tools exhibited a machined surface roughness $S_q$ parameter of 0.63 and 0.58 μm, respectively, at the machining distance of 10 m that decreased to 0.61 and 0.56 μm, respectively, at the machining distance of 20 m. This behavior is attributed to the consumption of asperities on the film's surface with the increase in machining distance and reduced ploughing effect. Subsequently, the machined surface roughness value rises with the increase in machining distance. Eventually, the a-C:H-coated tool failed at the machining distance of 82 m with the machined surface roughness $S_q$ parameter of 0.76 μm, while the a-C-coated tool failed at 121 m with the machined surface roughness $S_q$ parameter of 0.65 μm. The behavior of tool life may be attributed to the H atoms of the a-C:H coating passivating the C bond and showing a higher $sp^3$ bond and degree of disorder, but at the same time reducing the degree of cross-linking of the internal structure, which results in a decrease in hardness. Hence, the a-C:H-coated tool shows a lower service life than the a-C:H-coated tool [7]. In a word, both the tool life and machined surface roughness of the a-C:H and the a-C-coated tools are superior to those of the uncoated tool. The a-C film shows better machining performance than the a-C:H film.

The surface morphology of the 2A50 aluminum alloy workpieces, obtained with different cutting tools, was characterized by a white-light interferometer and is shown in Figure 4a–c. It illustrates that the surface morphologies of the workpiece machined by the a-C:H-coated tool, the a-C-coated tool, and the uncoated tool were significantly different in the stable cutting stage. The micrographs clearly indicated the interaction between the blade and the workpiece. The surface morphology machined by the a-C:H-coated tool is shown in Figure 4a. There are both shallow and deep grooves on the workpiece surface and the Z-axis depth of the deep groove is around 2.2 μm. There are only deep grooves on the surface machined by the uncoated cutting tool, as shown in Figure 4c, while only shallow grooves are observed on the surface machined by the a-C-coated tool, as shown in Figure 4b. Moreover, the Z-axis depth of the groove is up to 3.5 μm in Figure 4c, which is deeper than that in Figure 4a.

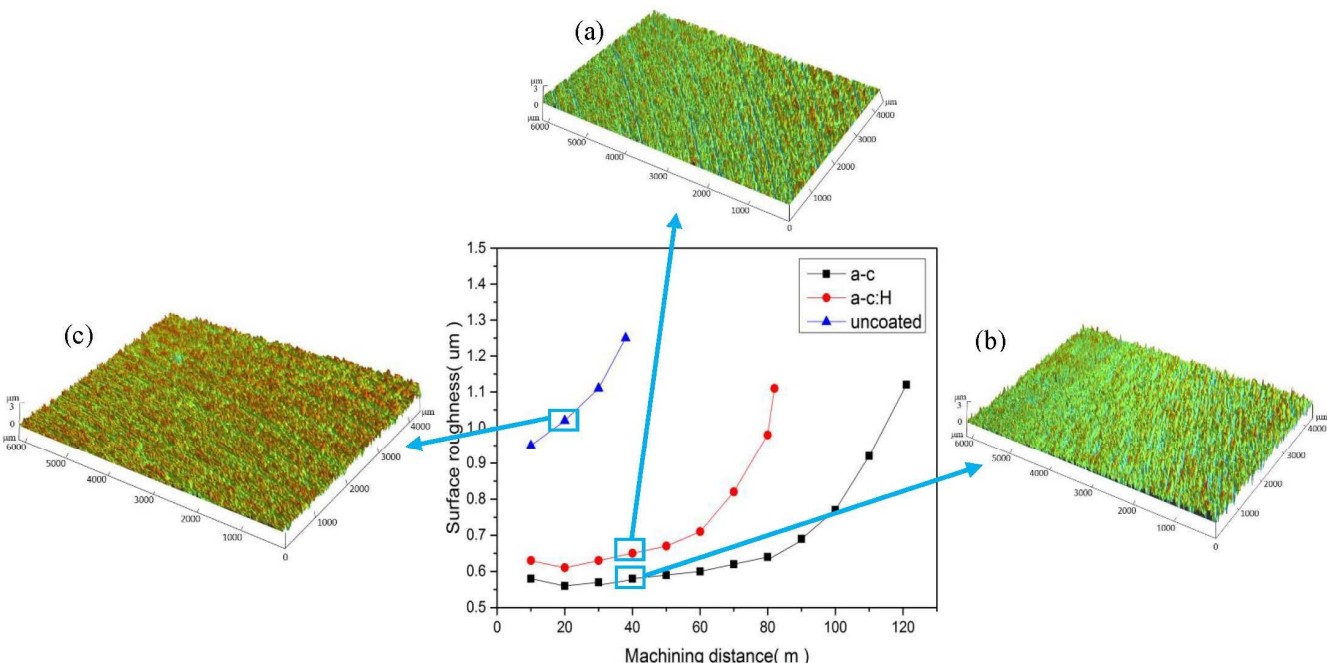

**Figure 4.** The relationship between machining distance and machined surface roughness: (**a**) a-C:H-coated tool, (**b**) a-C-coated tool, and (**c**) uncoated tool.

Optical photographs of tool wear and corresponding micrographs of chips in the stable cutting stage (the a-C and a-C:H-coated tools after cutting 40 m, and the uncoated tool after cutting 20 m) were obtained by an optical microscope, as shown in Figure 5. The chip from the uncoated tool has an irregular surface appearance, as shown in Figure 5c, while a build-up edge appears on the a-C:H-coated tool in Figure 5a. Only small wear occurred on the a-C-coated tool in Figure 5b, which indicates that the a-C:H and a-C coating could improve the anti-adhesion and service life of tools. The results are consistent with the research of Hanyu et al. [28], Fukui et al. [29], and Vandevelde et al. [30]. Moreover, the anti-adhesion and wear-resistance of the a-C-coated tool are superior to that of the a-C:H-coated tool from Figure 5a,b. The corresponding chips are shown in Figure 5d–f, where it can be seen that the chips are flat machined by the a-C:H-coated tool and several scratches are present in the flat chips in Figure 5d. Those behaviors may be attributed to the unstable cutting force and the variation coinciding with the fracturing cycle due to the build-up edge on the a-C:H-coated tool [30]. The shape of chips is curly from the a-C-coated tool, as shown in Figure 5e, and is either flat or spiral from the uncoated tool, as shown in Figure 5f. In addition, the surface of the curly chips from the a-C-coated tool machining is smooth while the surface of that from the uncoated tool machining is rough with obvious scratches. Meanwhile, it was observed from the infrared thermometer that the maximum temperature at the a-C:H-coated tool tip during cutting was 176 °C, while that of the a-C-coated tool was 141 °C. Hartley et al. [31], Mason et al. [32], and Potdar et al. [33] used the same method to measure the cutting temperature and other researchers found that the measured temperature was about 100 °C lower than the practical cutting temperature [34–36]. Therefore, the maximum temperature at the a-C:H and a-C-coated tool tip is 276 °C and 241 °C, respectively, when cutting aluminum alloy.

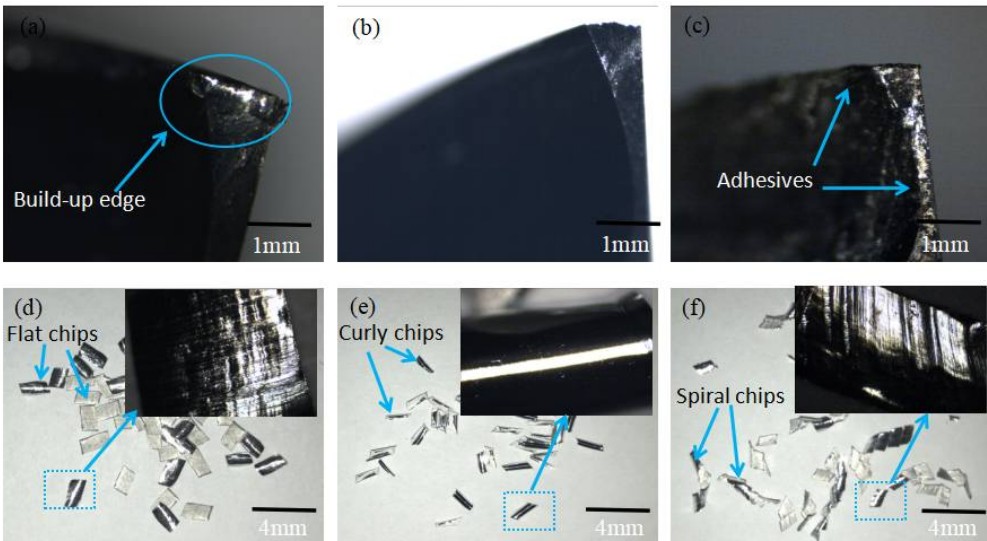

**Figure 5.** The tool wear optical photographs and corresponding micrographs of chips in the stable cutting stage: (**a**,**d**) a-C:H-coated tool and chips after cutting 40 m; (**b**,**e**) a-C-coated tool and chips after cutting 40 m, and (**c**,**f**) uncoated tool and chips after cutting 20 m.

## 4. Discussion

In this study, the cutting performance of the a-C:H and a-C-coated tools were investigated for dry cutting 2A50 aluminum alloy. The results reveal that both the a-C:H and a-C coatings can improve the cutting performance of a milling cutter, regarding factors such as service life, the surface quality of the workpiece, and the anti-adhesive properties. However, the cutting performance of the a-C-coated tool was superior to that of the a-C:H-coated tool, with the difference between these two coatings being the hydrogen element. In order to investigate the effect of hydrogen on the interaction between the coatings and workpiece, the molecular simulation technique was applied.

Interactions between Al/a-C:H and Al/a-C surfaces were simulated using first-principles calculations on the basis of density functional theory. The exchange correlation energy was calculated in the generalized gradient approximation by applying the Perdew–Burke–Emzerh functional form [37,38]. The Grimme method was used to perform density functional theory dispersion correction [39,40]. In addition, the ultra-soft pseudopotential [41] was used to describe the interaction between two surfaces. The 450 eV cutoff energy and $10 \times 10 \times 1$ k-point grids were adopted throughout this work. The convergence criteria during relaxations were selected as follows: $1.0 \times 10^{-5}$ eV/atom, 0.05 e V/Å, 0.1 Gpa, and $1.0 \times 10^{-3}$ Å for energy, maximum force, maximum stress, and maximum displacement, respectively. The calculated lattice parameters of Al and diamond were less than 1% of the experimental values [42]. The a-C:H and a-C-coated surfaces were represented by an H-terminated diamond surface (diamond:H) and a diamond surface, following the common practice used in the literature [43,44] of employing a diamond as the model to study the a-C:H and a-C surface. The convergence studies reveal that the use of six layers of diamond(111) and ten layers of Al(111) is sufficient for simulating the bulk effect in a surface slab. The first-principles molecular dynamics (FPMD) simulation was performed by employing the interface model, as shown in Figure 6, at the diamond:H(111)/Al(111) layer (40 aluminum atoms, 24 carbon atoms, and 4 hydrogen atoms) or the diamond(111)/Al(111) layer (40 aluminum atoms and 24 carbon atoms) and the vacuum layer of 15 Å. Less than 3% lattice mismatch occurred at the interface. Therefore, since the error of the periodic calculation of the interface structure is less than 5%, the effect of mismatch can be ignored [45]. The constant number of particles–volume–temperature ensemble was employed for FPMD simulations with the duration of 25 ps and the simulation temperature of the tow interface

model was maintained at 549 K (in order to reduce the influence of temperature on Al and diamond material) using a nose thermostat [46].

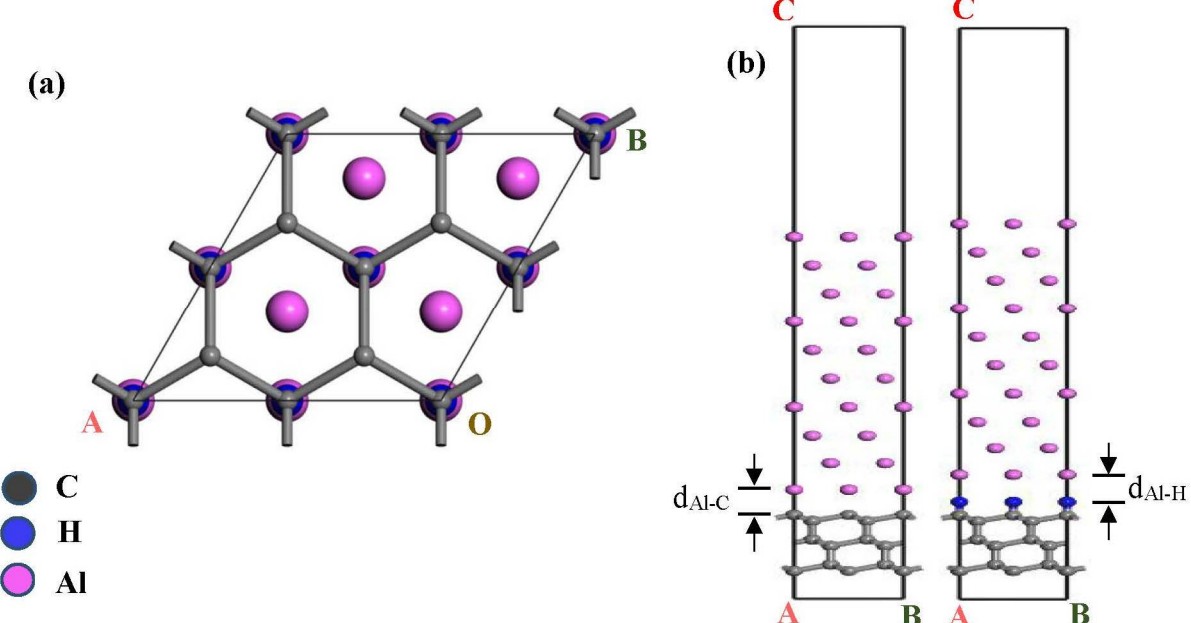

**Figure 6.** The interface model used in the first-principles calculations: (**a**) Top view of the interface registry, where the edge length of the cell is 5.05 Å. (**b**) Side view of the interface model formed between 10 layers of Al and 6 layers of diamond/diamond surface terminated with hydrogen, where $d_{Al-C}$ is the distance between the Al and C atoms and $d_{Al-H}$ is the distance between the Al and H atoms at the interface.

The work of adhesion ($W_{ad}$) is used to characterize interfacial bonding of the a-C:H/Al and a-C/Al interface. It can be calculated by the formula:

$$W_{ad} = (E_{slab1} + E_{slab2} - E_{interface})/A \qquad (1)$$

where $E_{slab1}$ denotes the total energy of the a-C:H(a-C) slab, $E_{slab2}$ denotes the total energy of an Al(111) slab with ten layers, and $E_{interface}$ denotes the total energy of the interface system. The total interface area is given by A.

The $W_{ad}$ was calculated using relaxed geometries. The total energies of optimal geometries with the interfacial distance of 1.81 Å for $d_{Al-H}$ and $d_{Al-C}$ were calculated in this paper. The interfacial distance and work of adhesion are listed in Table 2.

**Table 2.** The interfacial distance and work of adhesion.

| Interface | Interfacial Distance (Å) | | Work of Adhesion ($W_{ad}$) |
|---|---|---|---|
| | Unrelaxed | Fully Relaxed | |
| a-C:H/Al | 1.81 | 1.52 | 5.27 J/m$^2$ |
| a-C/Al | 1.81 | 1.74 | 4.21 J/m$^2$ |

The model with the larger $W_{ad}$ will be more stable, while the model shows smaller distances. As we can see from Table 2, the interfacial distance of the full relaxed interface system is 1.52 Å for a-C:H/Al and 1.74 Å for a-C/Al. The $W_{ad}$ of the a-C/Al interface is 4.21 J/m$^2$ and the $W_{ad}$ of the a-C:H/Al interface is 5.27 J/m$^2$. The larger $W_{ad}$ and smaller distances of the a-C:H/Al interface indicate that the anti-adhesion of the a-C:H-coated tool is weaker than that of the a-C-coated tool in cutting aluminum alloy. Meanwhile, the movement of the carbon atom is perpendicular to the interface in the a-C:H/Al interface model and a lateral movement of the H atoms is also observed in the FPMD simulation,

while the interfacial atoms of the a-C/Al model move only in the direction perpendicular to the interface. The lateral motion of the H atoms changes the charge distribution at the interface and leads to the increase in $W_{ad}$, which is also in agreement with the research of Jin et al. [47].

To further describe the large $W_{ad}$ behavior of the a-C:H/Al model, the model is compared before and after optimization, as shown in Figure 7. In addition, the bonding strength between C–H bonds is calculated by population analysis. The overlap population may be used to assess the covalent or ionic nature of a bond. A high value of the bond population indicates a covalent bond, while a low value of the bond population indicates an ionic interaction [48,49]. It can be seen from Figure 7 that the distance and angle of the C–H bonds increase from 1.14 Å to 1.348 Å and from 90° to 123.456° after optimization. The distance between the C atoms and Al atoms near the interface decreases from 2.95 Å to 2.482 Å as the position of H atoms changes. Furthermore, the distance of carbon atoms is 1.348 Å at the second layer and it increases to 1.632 Å at the third layer. Meanwhile, the population of C–H bonds decreases from 0.63 to 0.54.

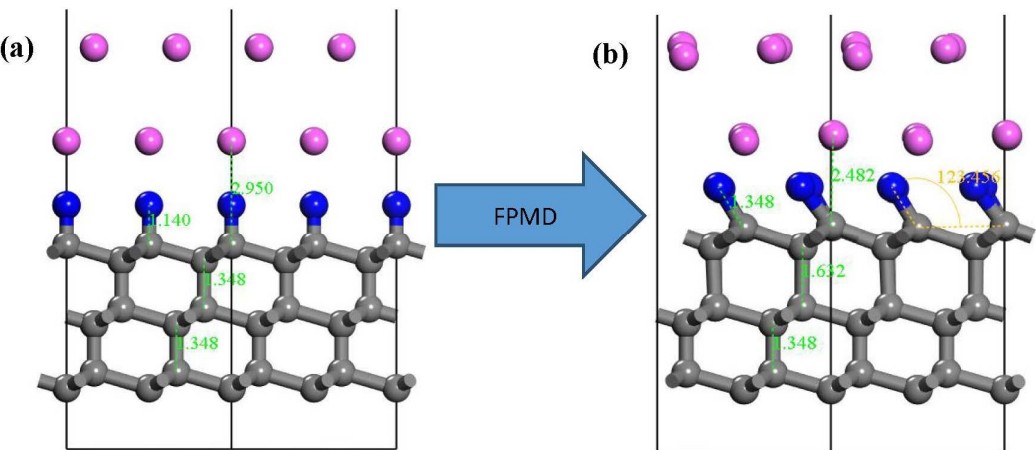

**Figure 7.** The a-C/Al interface model: (**a**) initial model and (**b**) optimized model.

The behavior of the C atoms at the interface may be attributed to the horizontal motion of H atoms during the FPMD simulation, which leads to a change in the charge distribution of C atoms. The fracture trend of the C–H bonds could be obtained from the bond length elongation and the population reduction. Previous research has shown that the C–H bond breaks above 300 °C because the hydrogen atoms are desorbed from the DLC film, leaving many dangling bonds and increasing the attraction of the film to other materials [7–9]. Therefore, it can be concluded that H atoms in the a-C:H coating will have transverse motion below 300 °C, which will change the charge distribution of C atoms at the interface and increase the interaction between C and Al. It is easier for aluminum alloys to adhere to the a-C:H-coated tools than to the a-C-coated tools in the machining process. At the same time, the position of other carbon atoms in the coating will also change. However, the suspension bond that emerged interacts more strongly with Al alloy when C–H bonds break. All of the above data can explain why the a-C coating shows better adhesion resistance than the a-C:H coating when applied to aluminum alloy cutting.

## 5. Conclusions

The machining behaviors of the a-C and a-C:H-coated tools when cutting 2A50 aluminum alloy were investigated in this paper. The anti-adhesion mechanisms were revealed using first-principles molecular dynamics simulation. The as-deposited a-C:H and a-C coatings can improve tool performance when cutting aluminum alloys. However, the improvement of tool performance is related to the adhesion resistance of the coating. A longer service life of 121 m and better surface quality ($S_q$ parameter of 0.23 μm) can be obtained by using an a-C coating with superior anti-adhesion. The a-C:H coating has a

higher interaction with Al because the horizontal motion of H atoms can lead to changes in the charge distribution of C atoms at the interface before the C–H bond breaks. Hence, the adhesion resistance of the a-C:H coating is reduced.

**Author Contributions:** B.H. and Q.Z. conceived, designed and performed the experiments; B.H. and E.-g.Z. analyzed the date and wrote the paper; R.-c.L. and H.-m.D. provided materials/analysis tools. All authors have read and agreed to the published version of the manuscript.

**Funding:** This research was financially supported by the Natural Science Foundation of China (Grant No. 51971148), Natural Science Foundation of Shanghai (Grant No. 20ZR1455700), Key Support Plan of Xiamen Science and Technology Committee (Grant No.3502Z20191022), and Science and Technology Commission of Shanghai Municipality (Grant No. 20DZ2303300).

**Data Availability Statement:** Data available in a publicly accessible repository.

**Conflicts of Interest:** The authors declare no conflict of interest.

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
