# Peer review of "Research on the Performance of Diamond-Like Carbon Coatings on Cutting Aluminum Alloy: Cutting Experiments and First-Principles Calculations"

_coatings, doi:10.3390/coatings11010063_

Round 1

Reviewer 1 Report

The authors wrote a very interesting article that could be even better with the following corrections:

  1. Line 22: the sentence fragment reads as follows "workpiece surface roughness Rq" statement will sounded better to follow for example: root mean square deviation Rq or Rq parameter, which is used to describe the surface roughness. Please include a note throughout the article (Line 143,193,195,202,).
  2. Rq parameter, i.e. a parameter in the 2D system, has been determined, and in Figure 1, data can be observed in the 3D system, it is worth using parameters to describe surface roughness in the 3D system, i.e. Sa, St or Sq. R parameters are used when the data is in the form of a roughness profile, if we have data in the form of a surface, it is worth using S parameters to describe the surface roughness, S parameters give more complete information.
  3. The results in subsections 3.1 and 3.2 are very briefly described.

Author Response

Dear reviewer, Happy New Year.

  1. Line 22: the sentence fragment reads as follows "workpiece surface roughness Rq" statement will sounded better to follow for example: root mean square deviation Rqor Rq parameter, which is used to describe the surface roughness. Please include a note throughout the article (Line 143,193,195,202,).

Answer: Thank you very much for your suggestion. It has been modified as “workpiece surface roughness Sq parameter” in the submitted version

  1. Rqparameter, i.e. a parameter in the 2D system, has been determined, and in Figure 1, data can be observed in the 3D system, it is worth using parameters to describe surface roughness in the 3D system, i.e. Sa, St or Sq. R parameters are used when the data is in the form of a roughness profile, if we have data in the form of a surface, it is worth using S parameters to describe the surface roughness, S parameters give more complete information.

Answer: Sq. It has been revised in submitted manuscript.

3.The results in subsections 3.1 and 3.2 are very briefly described.

Answer:It has been revised in the submitted manuscript.

.

Reviewer 2 Report

The authors present a careful method to interpret the benefit of using an a-C coating vs un-coated mills by investigating changes in interface properties with or without Hydrogen.

The experimental results are really interesting, but the demonstration needs caution.

Why did the authors chose only the 111 / 111 interfaces?.

By ignoring the mismatch between the two surfaces (line 290), You remove important physical phenomena which could be grasped by the calculations, namely considering a perfect fit between diamond and Al drastically increases the bond between the two surfaces (forming a pin). Therefore, only having C atoms on top of Al atoms will increase the bond and can artificially remove any dislocation forming effects due to the real mismatch. Therefore, the interpretation of results needs to be made with caution. One way to verify how this can be neglected is by performing a calculation with the correct positioning (despite the assumption of the authors to neglect it).

The conclusion that H atoms in the coating will have transverse motion is drawn from papers from other authors (reference 9). Another way of verifying this is by modifying the temperature in the MD thermostat and check whether it is the case.

These suggestions could greatly enhance the discussion part of the paper.

Few corrections

The manuscript received for review contained an important amount of previous track changes which made the read more difficult as if the manuscript was not finished and was in draft mode.

Formatting problem on figure 5. Arrows are offset as well as (a) to (f) and the pictures. Similar offset to the picture is present in Fig 4 (additional formatting checks are required throughout)

Line 366 better adhesion resistance

Line 381 because the horizontal

Author Response

Dear reviewer, Happy New Year.

  1. Why did the authors chose only the 111 / 111 interfaces?.

Answer:Since the low index surface of diamond and aluminum are both 111 surfaces, the interaction between two low index surfaces is stronger than that between other surfaces. Therefore, the surface with the strongest interaction is selected for the simulation.

  1. By ignoring the mismatch between the two surfaces (line 290), You remove important physical phenomena which could be grasped by the calculations, namely considering a perfect fit between diamond and Al drastically increases the bond between the two surfaces (forming a pin). Therefore, only having C atoms on top of Al atoms will increase the bond and can artificially remove any dislocation forming effects due to the real mismatch. Therefore, the interpretation of results needs to be made with caution. One way to verify how this can be neglected is by performing a calculation with the correct positioning (despite the assumption of the authors to neglect it).

Answer:The mismatch will affect the interaction between diamond and aluminum, but 3 percent of mismatch can be ignored, which has very little influence on the results. It can be seen from the literature (for example, References 45) that a mismatch degree of less than 5% can be ignored, because it has little impact on the calculation results.

  1. The conclusion that H atoms in the coating will have transverse motion is drawn from papers from other authors (reference 9). Another way of verifying this is by modifying the temperature in the MD thermostat and check whether it is the case.

Answer:Temperature affects the motion of H atom. In this paper, it is the cutting heat that causes the vibration of C-H bond, so that H atom will have transverse motion.

4.Few corrections

The manuscript received for review contained an important amount of previous track changes which made the read more difficult as if the manuscript was not finished and was in draft mode.

Formatting problem on figure 5. Arrows are offset as well as (a) to (f) and the pictures. Similar offset to the picture is present in Fig 4 (additional formatting checks are required throughout)

Line 366 better adhesion resistance

Line 381 because the horizontal

Answer: Thank you very much for your suggestion. We have made corrections in the submitted manuscript.

This manuscript is a resubmission of an earlier submission. The following is a list of the peer review reports and author responses from that submission.

Round 1

Reviewer 1 Report

Many comments are shown in the attached file.

However, in my opinion, the quality of this manuscript could be improved. 

In particular, I refer to the description of the investigated samples (Ti, Ti-C and WC???), the thickness of the coatings, the Raman spectroscopy investigation.  Finally, I think that it should be important to characterize the coatings by indentation or nanoindentation tests.

Reviewer 2 Report

The authors wrote a very interesting article that could be even better with the following corrections:

  1. Line 18 and 19: What does it mean "(average roughness of 0.23um)". Normalized parameters are used to describe the surface roughness.
  2. Line 118-120: The authors use the RMS description, probably the Rq parameter. Please let me know which parameter to describe the surface roughness was determined and according to which standard. If the Rq parameter, i.e. a parameter in the 2D system, has been determined, and in Figure 1, data can be observed in the 3D system, it is worth using parameters to describe surface roughness in the 3D system, i.e. Sa, St or Sq.
  3. Line 118: you cannot describe the surface in this way, "surface roughness RMS of 30nm" I suggest. The Rq parameter to describe the surface roughness is 30 nm. The surface roughness cannot be described by numbers, this is what the parameters for assessing the surface roughness are for. Please edit descriptions throughout the article.
  4. The results in subsections 3.1 and 3.2 are very briefly described.
  5. Line 158, Fig.4, Line 19, Line 160, Line 165, Line 166...The surface roughness cannot be described by numbers, this is what the parameters for assessing the surface roughness are for.
  6. Chapter 2 (should be marked as 4) does not have the characteristics of a Discussion.
  7. Chapter 5 The conclusions are poorly described.